# COLLABORATIVE GENERATED HASHING FOR MARKET ANALYSIS AND FAST COLD-START RECOMMENDATION

## ABSTRACT

Cold-start and efficiency issues of the Top-k recommendation are critical to large-scale recommender systems. Previous hybrid recommendation methods are effective to deal with the cold-start issues by extracting real latent factors of cold-start items(users) from side information, but they still suffer low efficiency in online recommendation caused by the expensive similarity search in real latent space. This paper presents a collaborative generated hashing (CGH) to improve the efficiency by denoting users and items as binary codes, which applies to various settings: cold-start users, cold-start items and warm-start ones. Specifically, CGH is designed to learn hash functions of users and items through the Minimum Description Length (MDL) principle; thus, it can deal with various recommendation settings. In addition, CGH initiates a new marketing strategy through mining potential users by a generative step. To reconstruct effective users, the MDL principle is used to learn compact and informative binary codes from the content data. Extensive experiments on two public datasets show the advantages for recommendations in various settings over competing baselines and analyze the feasibility of the application in marketing.

## 1 INTRODUCTION

With the explosion of e-commerce, most customers are accustomed to receiving a variety of recommendations, such as movies, books, news, or hotels they might be interested in. Traditional recommender systems just recommended items that are similar to what they liked or rated in the previous. Recommendations help users find their desirable items, and also creates new revenue opportunities for vendors, such as Amazon, Taobao, eBay, etc. Among them, one of the most popular recommendation methods, collaborative filtering is dependent on a large amount of user-item interactive information to provide an accurate recommendation. However, most of new e-commerce vendors do not have enough interactive data, which leads to low recommendation accuracy, i.e., cold-start issues.

Previous studies on cold-start issues generally modeled as a combination of collaborative filtering and content filtering, known as hybrid recommender systems. Specifically, they learned real latent factors by incorporating the side information into the interactive data. Such as Collaborative Deep Learning (CDL) (Wang et al., 2015), Visual Bayesian Personalized Ranking (VBPR) (He & McAuley, 2016), Collaborative Topic modeling for Recommedation (CTR) (Wang & Blei, 2011), and the DropoutNet for addressing cold start (DropoutNet)(Volkovs et al., 2017), ABCPRec for Bridging Consumer and Producer Roles for User-Generated Content Recommendation (ABCPRec)(Tsukuda et al., 2019). All of the above hybrid recommender systems were modeled in real latent space, which leads to low efficiency for the online recommendation with the increasing scale of datasets.

Recent studies show the promising of hashing based methods to tackle the efficiency challenge by representing users and items with binary codes (Zhang et al., 2014; Zhou & Zha, 2012; Zhang et al., 2016; Liu et al., 2019), because the preference score can be represented by the Hamming distance calculated via XOR operation efficient (Wang et al., 2014). However, the existing hashing based recommendations are learning-based frameworks, which leads to NP-hard problems of optimizing

discrete objectives. Thus many scholars learned binary codes by some approximate techniques, such as the two-stage hashing learning method utilized in Preference Preserving Hashing(PPH) (Zhang et al., 2014) and the Iterative Quantization(ITQ) (Zhou & Zha, 2012). To reduce information loss, two learning-based hashing frameworks: bit-wise learning and block-wise learning were respectively proposed in hashing based recommendation frameworks (Zhang et al., 2016; Wang et al., 2019; Zhang et al., 2018; Zheng et al.).

However, due to the requirement of binary outputs for learning-based hashing frameworks, the training procedure is expensive for large-scale recommendation, which motivates us to propose a generative approach to learn hash functions. In this paper, we propose the collaborative generated hashing(CGH) to learn hash functions of users and items from content data with the principle of Minimum Description Length (MDL) (Dai et al., 2017).

In marketing area, mining potential customers is crucial to the e-commerce. CGH provides a strategy to discover potential users by the generative step. To reconstruct effective users, uncorrelated and balanced limits are imposed to learn compact and informative binary codes with the principle of the MDL. Especially, discovering potential customers is vital to the success of adding new items for a recommendation platform (Papies et al., 2017). Specifically, for a new item, we can generate a new potential user by the generative step (detailed in Section 2.1), and then search the nearest potential users in the user set. By recommending a new product to the potential users who might be interested in but didn't plan to buy, further e-commerce strategies can be developed to attract those potential users.

We organize the paper as follows: Section 2 introduce the main techniques of CGH. We first introduce the framework of CGH and compare it with the closely related competing baselines: CDL (Wang et al., 2015) and DropoutNet (Volkovs et al., 2017); we then formulate the generative step in Section 2.1 and the inference step in Section 2.2, respectively; we finally summarize the training objective and introduce the optimization in Section 2.3. Particularly, we demonstrate the process of mining potential users for the marketing application in Section 2.1. Section 3 presents the experimental results for marketing analysis and recommendation accuracy in various settings. Section 4 concludes the paper.

The main contributions of this paper are summarized as follows:

(1) We propose the Collaborative Generated Hashing (CGH) with the principle of MDL to learn compact but informative hash codes, which applies to various settings for recommendation.

(2) We provides a marketing strategy by discovering potential users by the generative step of CGH, which can be applied to boost the e-commence development.

(3) We evaluate the effectiveness of the proposed CGH compared with the state-of-the-art baselines, and demonstrate its robustness and convergence properties on the public datasets.

## 2 COLLABORATIVE GENERATED HASHING

The framework of the proposed CGH is shown in Fig. 1(c), where $U$, $V$ and $R$ are respectively observed user content, item content and rating matrix. $B$ and $D$ are binary codes of users and items, respectively. CGH consists of the generative step marked as dashed lines and the inference step denoted by solid lines. Once training is finished, we fix the model and make forward passes to obtain binary codes $B$ and $D$ through the inference step, and then conduct recommendation. For the marketing application, we create a new user via the generative step.

In comparison of CGH with the closely related baseline CDL (Wang et al., 2015), the proposed CGH aims to learn binary codes instead of real latent vectors $P$ and $Q$ due to the advantage of hashing for online recommendation; plus the CGH optimizes an objective with the principle of MDL, while CDL optimized the joint objective of rating loss and item content reconstruction error. In comparison of CGH with DropoutNet (Volkovs et al., 2017), CGH can be used as a marketing strategy by discovering potential users; plus CGH learns hash functions by stacked denoising autoendoer, while DropoutNet obtained real latent factors by the standard neural network.

In the following we start by first formulating the generative process and demonstrating the application in marketing area; we then formulate the inference step; we finally summarize the training objective and the optimization method.

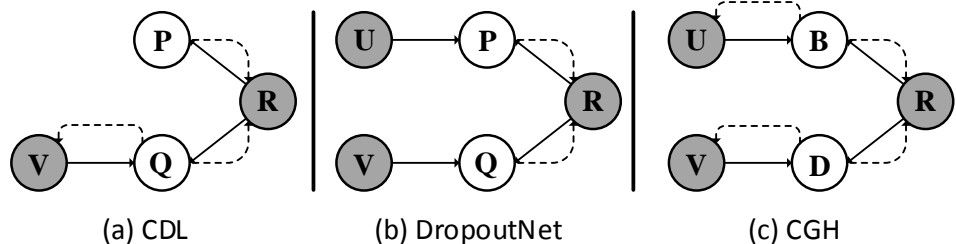

(a) CDL        (b) DropoutNet        (c) CGH

Figure 1: Differences between CDL, DropoutNet and our proposed CGH. Solid lines and dashed lines respectively represent the inference (encoding) and the generative (decoding) process. The shaded nodes $U, V, R$ are observed user content, item content and rating, respectively. $P(B)$, $Q(D)$ denotes real latent factors(binary codes) of users and items.

## 2.1 MINING POTENTIAL USERS

. Give a sparse rating matrix $R$ and item content data $V \in \mathbb{R}^{d_v}$, where $d_v$ is the dimension of the content vector, and $V$ is stacked by the bag-of-words vectors of item content in the item set $V$. most previous studies were focus on modeling deterministic frameworks to learn representations of items for item recommendation, such as CDL, CTR, DropoutNet, et.al. In this paper, we discover a new strategy from a perspective of marketing for item recommendation – mining potential users.

We demonstrate the process of mining potential users by an item through the generative step in Fig. 2. After the inference step, the binary code of item $j$ is available. By maximizing the similarity function $\delta(\boldsymbol{b}_i, \boldsymbol{d}_j)$ (detailed in Section 2.1), the optimal binary code $\boldsymbol{b}_p$ is obtained. Then we generate the new user $\boldsymbol{u}_p$ through the generative step. Finally we find out potential users from the user set by some nearest neighborhood algorithms, such as KNN. As a marketing strategy, it can discover potential users for both warm-start items and cold-start items. Thus, from a perspective of marketing, it can be regarded as another kind of item recommendation.

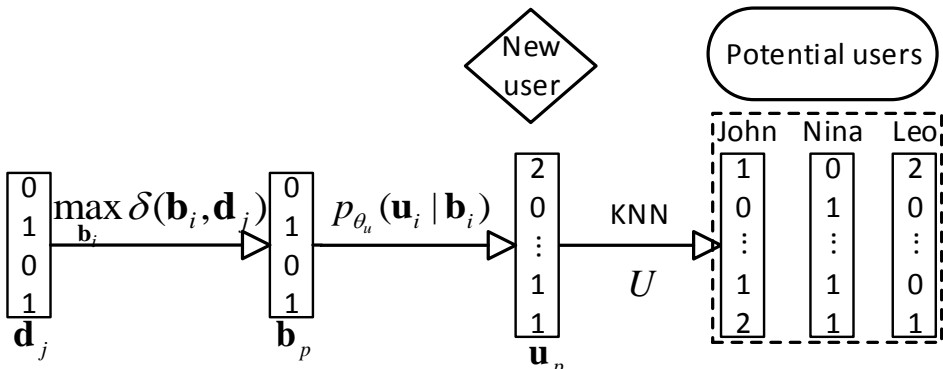

Figure 2: Demonstration of mining potential users for an item $j$. After the inference step, $\boldsymbol{d}_j$ is available, we first find out the most similar binary code $\boldsymbol{b}_p$; we then generate a new potential user $\boldsymbol{u}_p$ by the generative process; we furtherly search the top-$k$ nearest potential users from the user set by some nearest neighborhood algorithms (e.g., KNN).

The generation process (also referred as decoding process) is denoted by dashed lines in Fig. 1 (c). Fix binary codes $\boldsymbol{b}_i$ and $\boldsymbol{d}_j$ of the user $i$ and the item $j$, the bag-of-words vector $\boldsymbol{u}_i$ of the user $i$ ($\boldsymbol{v}_j$ of the item $j$) is generated via $p(\boldsymbol{\theta}_u)$. The ratings $r_{ij}$ is generated by $\boldsymbol{b}_i$ and $\boldsymbol{d}_j$. We use a simple Gaussian distribution to model the generation of $\boldsymbol{u}_i$ and $\boldsymbol{v}_j$ given $\boldsymbol{b}_i$ and $\boldsymbol{d}_j$ like Stochastic Generative Hashing (SGH) (Dai et al., 2017):

$$p(\boldsymbol{u}_i|\boldsymbol{b}_i) = \mathcal{N}(\mathcal{C}_u \boldsymbol{b}_i, \lambda_u^{-1} \boldsymbol{I}),$$
$$p(\boldsymbol{v}_j|\boldsymbol{d}_j) = \mathcal{N}(\mathcal{C}_v \boldsymbol{d}_j, \lambda_v^{-1} \boldsymbol{I}), \tag{1}$$

where $\mathcal{C}_{uk} = [\boldsymbol{c}_{uk}]_{k=1}^r, \boldsymbol{c}_{uk} \in \mathbb{R}^{d_u}$ is the codebook (Dai et al., 2017) with $r$ codewords, which is similar for $\mathcal{C}_v$, and $d_u$ is the dimension of the bag-of-the-words vector of users. The prior is modeled as the multivariate Bernoulli distribution on hash codes: $p(\boldsymbol{b}_i) \sim \mathcal{B}(\rho_u)$, and $p(\boldsymbol{d}_j) \sim \mathcal{B}(\rho_v)$, thus the prior probability are as follows:

$$
\begin{aligned}
p(\boldsymbol{b}_i) &= \prod\nolimits_{k=1}^r \rho_{ui}^{b_{ik}} (1 - \rho_{ui})^{1-b_{ik}}, \\
p(\boldsymbol{d}_j) &= \prod\nolimits_{k=1}^r \rho_{vj}^{d_{jk}} (1 - \rho_{vj})^{1-d_{jk}},
\end{aligned}
\tag{2}
$$

We formulate the rating with the similarity between binary codes of users and items like the most successful recommender systems, matrix factorization (Koren et al., 2009). Then the rating is thus drawn from the normal distribution centered at the similarity value,

$$
p(r_{ij}|\boldsymbol{b}_i, \boldsymbol{d}_j) \sim \mathcal{N}(\delta(\boldsymbol{b}_i, \boldsymbol{d}_j), C_{ij}^{-1}),
\tag{3}
$$

where $\delta(\boldsymbol{b}_i, \boldsymbol{d}_j) = 1 - \frac{1}{r}\text{Hamdis}(\boldsymbol{b}_i, \boldsymbol{d}_j)$ denotes the similarity between binary codes $\boldsymbol{b}_i$ and $\boldsymbol{d}_j$. Hamdis$(\boldsymbol{b}_i, \boldsymbol{d}_j)$ represents Hamming distance between the two binary vectors, which has been widely applied in hashing-based recommender system (Wang & Blei, 2011; Lian et al., 2017; Zhang et al., 2018). $C_{ij}$ is the precision parameter that serves as confidence for $r_{ij}$ similar to that in CTR (Wang & Blei, 2011) ($C_{ij} = a$ if $r_{ij} = 1$ and $C_{ij} = b$ otherwise) due to the fact that $r_{ij} = 0$ means the user $i$ is either not interested in item $j$ or not aware of it.

With the generative model constructed, the joint probability of both observed ratings, content vectors and binary codes is given by

$$
p\left(\boldsymbol{R}, \boldsymbol{U}, \boldsymbol{V}, \boldsymbol{B}, \boldsymbol{D}\right) = \prod_{i,j} p(r_{ij}|\boldsymbol{b}_i, \boldsymbol{d}_j)p(\boldsymbol{u}_i|\boldsymbol{b}_i)p(\boldsymbol{v}_j|\boldsymbol{d}_j)p(\boldsymbol{b}_i)p(\boldsymbol{d}_j)
\tag{4}
$$

## 2.2 Constraints on Binary Latent Variables

The inference process (also referred as encoding process) shown in the Fig. 1 (c) with dashed lines, the binary latent variables $\boldsymbol{b}_i$ ($\boldsymbol{d}_j$) depends on the content vector $\boldsymbol{u}_i$ ($\boldsymbol{v}_j$) and the rating $\boldsymbol{R}$ (shadowed in Fig 1). Inspired by the recent work on generative hashing (Dai et al., 2017) and DropoutNet (Volkovs et al., 2017), we use a multivariate Bernoulli distribution to model the inference process of $\boldsymbol{b}_i$ and $\boldsymbol{d}_j$ with linear parametrization, i.e.,

$$
\begin{aligned}
q(\boldsymbol{b}_i|\tilde{\boldsymbol{u}}_i) &= \mathcal{B}(\sigma(\mathcal{T}_u^T\tilde{\boldsymbol{u}}_i)) \\
q(\boldsymbol{d}_j|\tilde{\boldsymbol{v}}_j) &= \mathcal{B}(\sigma(\mathcal{T}_v^T\tilde{\boldsymbol{v}}_j)),
\end{aligned}
\tag{5}
$$

where $\tilde{\boldsymbol{u}}_i = [\boldsymbol{u}_i, \boldsymbol{p}_i]$, $\tilde{\boldsymbol{v}}_j = [\boldsymbol{v}_j, \boldsymbol{q}_j]$. $\boldsymbol{p}_i$ and $\boldsymbol{q}_j$ are the results of $r$-dimension matrix factorization (Koren et al., 2009) of $\boldsymbol{R}$, i.e., $r_{ij} \approx \boldsymbol{p}_i^T\boldsymbol{q}_j$. $\mathcal{T}_u = [\boldsymbol{t}_{uk}]_{k=1}^r, \boldsymbol{t}_{uk} \in \mathbb{R}^{d_u+r}, \mathcal{T}_v = [\boldsymbol{t}_{vk}]_{k=1}^r, \boldsymbol{t}_{vk} \in \mathbb{R}^{d_v+r}$ are the transformation matrices of linear parametrization. From SGH (Dai et al., 2017), the MAP solution of the eq. (5) is readily given by

$$
\begin{aligned}
\boldsymbol{b}_i &= \underset{\boldsymbol{b}_i}{\text{argmax}}\, q(\boldsymbol{b}_i|\boldsymbol{u}_i) = \frac{\text{sign}(\mathcal{T}_u^T\boldsymbol{u}_i) + 1}{2}, \\
\boldsymbol{d}_j &= \underset{\boldsymbol{d}_j}{\text{argmax}}\, q(\boldsymbol{d}_j|\boldsymbol{v}_j) = \frac{\text{sign}(\mathcal{T}_v^T\boldsymbol{v}_j) + 1}{2}.
\end{aligned}
\tag{6}
$$

With the linear projection followed by a sign function, we can easily get hash codes of users and items. However, hashing with a simple sign function suffers from large information loss according to (Zhang et al., 2016), which motivates us to add constraints on parameters in the inference step.

To derive compact and informative hash codes for users and items, we add balanced and uncorrelated constraints in the inference step. The balanced constraint is proposed to maximize the entropy of each binary bit (Zhou & Zha, 2012), and the uncorrelated constraint makes each bit is independent of others. Then we can obtain compact and informative hash codes by the following constraints,

$$
\text{Balanced constraint: } \sum_k b_{ik} = 0, \sum_k d_{jk} = 0
\tag{7}
$$
$$
\text{Uncorrelated constraint: } \boldsymbol{b}_i\boldsymbol{b}_i^T = \boldsymbol{I}_r, \boldsymbol{d}_j\boldsymbol{d}_j^T = \boldsymbol{I}_r
$$

From the eq. (6), $\boldsymbol{b}_i$ and $\boldsymbol{d}_j$ are only dependent on parameters $\mathcal{T}_u$ and $\mathcal{T}_v$, respectively, thus we add constraints on $\mathcal{T}_u$ and $\mathcal{T}_v$ directly. So eq. (7) is equivalent to the following constraints,

$$\text{Balanced constraint: } \mathcal{T}_u^T \mathbf{1}_r = \mathbf{0}, \mathcal{T}_v^T \mathbf{1}_r = \mathbf{0},$$
$$\text{Uncorrelated constraint: } \mathcal{T}_u^T \mathcal{T}_u = \boldsymbol{I}_{d_u+r}, \mathcal{T}_v^T \mathcal{T}_v = \boldsymbol{I}_{d_v+r}. \tag{8}$$

By imposing the above constraints in the training step, compact and informative hash codes can be obtained through the inference process. Next we summarize the training objective and its optimization.

## 2.3 TRAINING OF CGH

Since our goal is to reconstruct users, items and ratings by using the least information of binary codes, we train the CGH with the MDL principle, which finds the best parameters that maximally compress the training data and meanwhile keep the information carried, thus CGH aims to minimize the expected amount of informations related to $q$:

$$\begin{aligned}
\mathcal{L}(q) =& E_q[\log p(\boldsymbol{R}, \boldsymbol{U}, \boldsymbol{V}, \boldsymbol{B}, \boldsymbol{D}) - \log q(\boldsymbol{B}, \boldsymbol{D})] \\
=& E_q[\log p(\boldsymbol{R}|\boldsymbol{B}, \boldsymbol{D}) + \log p(\boldsymbol{U}|\boldsymbol{B}) + \log p(\boldsymbol{V}|\boldsymbol{D}) - KL(q(\boldsymbol{B}|\tilde{\boldsymbol{U}})||p(\boldsymbol{B}))- \\
& KL(q(\boldsymbol{D}|\tilde{\boldsymbol{V}})||p(\boldsymbol{D}))]
\end{aligned} \tag{9}$$

Maximizing the posterior probability is equivalent to maximizing $\mathcal{L}(q)$ by only considering the variational distribution of $q(\boldsymbol{B}, \boldsymbol{D})$, the objective becomes

$$\begin{aligned}
\mathcal{L}_{MAP}(\Theta, \Phi) = & - \sum_{i,j} \frac{C_{ij}}{2} (r_{ij} - \delta(\boldsymbol{b}_i, \boldsymbol{d}_j))^2 - \frac{\lambda_u}{2} \sum_i (u_i - \mathcal{C}_v \boldsymbol{b}_i)^2 - \frac{\lambda_v}{2} \sum_j (\boldsymbol{v}_j - \mathcal{C}_v \boldsymbol{d}_j)^2 \\
& - \text{KL}(q_{\boldsymbol{\phi}_u}||p_{\boldsymbol{\theta}_u}) - \text{KL}(q_{\boldsymbol{\phi}_v}||p_{\boldsymbol{\theta}_v}) - \nabla(\Theta, \Phi)
\end{aligned} \tag{10}$$

where $\Theta = \{\boldsymbol{\theta}_u, \boldsymbol{\theta}_v\}$, $\Phi = \{\boldsymbol{\phi}_u, \boldsymbol{\phi}_v\}$, $\nabla(\Theta, \Phi)$ is the regularizer term with parameters $\Theta$ and $\Phi$. By training the objective in eq. (10), we obtain binary codes, but some bits probably be correlated. To minimize the reconstruction error, SGH had to set up the code length as long as $r = 200$. Our goal in this paper is to obtain compact and informative hash codes, thus we impose the balance and independent constraints on hash codes by eq. (8). Maximizing the eq. (10) is transformed to minimizing the following constrained objective function of the proposed Collaborative Generative Hashing (CGH),

$$\begin{aligned}
\mathcal{L}_{CGH}(\Theta, \Phi) = & \sum_{i,j} \frac{C_{ij}}{2} (r_{ij} - \delta(\boldsymbol{b}_i, \boldsymbol{d}_j))^2 + \frac{\lambda_u}{2} \sum_i (u_i - \mathcal{C}_u \boldsymbol{b}_i)^2 + \frac{\lambda_v}{2} \sum_j (\boldsymbol{v}_j - \mathcal{C}v \boldsymbol{d}_j)^2 + \\
& \text{KL}(q_{\boldsymbol{\phi}_u}||p_{\boldsymbol{\theta}_u}) + \text{KL}(q_{\boldsymbol{\phi}_v}||p_{\boldsymbol{\theta}_v}) + \alpha_u \left\|\mathcal{T}_u^T \mathbf{1}_r\right\|_2^2 + \alpha_v \left\|\mathcal{T}_v^T \mathbf{1}_r\right\|_2^2 + \\
& \beta_u \left\|\mathcal{T}_u^T \mathcal{T}_u - \boldsymbol{I}_{d_u+r}\right\|_2^2 + \beta_v \left\|\mathcal{T}_v^T \mathcal{T}_v - \boldsymbol{I}_{d_v+r}\right\|_2^2 + \nabla(\Theta, \Phi).
\end{aligned} \tag{11}$$

The objective of CGH in eq. (11) is a discrete optimization problem, which is difficult to optimize straightforwardly, so in the training stage, the tanh function is utilized to replace the sign function in the inference step, and then the continuous outputs are used as a relaxation of hash codes.

With the relaxation, we train all components jointly with back-propagation. After training, we fix them and make forward passes to map the concatenate vectors in $\tilde{U}$ and $\tilde{V}$ to binary codes $\boldsymbol{B}$ and $\boldsymbol{D}$, respectively. The recommendation in various settings is then conducted using $\boldsymbol{B}$ and $\boldsymbol{D}$ by the similarity score estimated as before $\delta(\boldsymbol{b}_i, \boldsymbol{d}_j) = 1 - \frac{1}{r}\text{Hamdis}(\boldsymbol{b}_i, \boldsymbol{d}_j)$.

The training settings are dependent on the recommendation settings, i.e, warm-start, cold-start item, and cold-start user. $\mathcal{L}_{CGH}(\Theta, \Phi)$ aims to minimize the rating loss and two content reconstruction errors with regularizers. (a.) For the warm-start recommendation, ratings for all users and items are available, then the above objective is trivially optimized by setting the content weights to 0 and learning hashing function with the observed ratings $\boldsymbol{R}$. (b.) For the cold-start item recommendation, ratings for some items are missing, then the objective is optimized by setting the user content weight to 0 and learning parameters with the observed ratings $\boldsymbol{R}$ and item content $\boldsymbol{V}$. (c.) The training setting for the cold-start user recommendation is similar to the cold-start item recommendation.

## 3 EXPERIMENTS

We validate the proposed CGH on two public dataset: CiteUlike[1] and RecSys 2017 Challenge dataset[2] from the following two aspects:

(1) **Marketing analysis.** To validate the effectiveness of CGH in marketing area, we fist defined a metric to evaluate the accuracy of mining potential users; we then test the performance for warm-start item and cold-start item, respectively.

(2) **Recommendation performance.** We test the performance of CGH for recommendation in various settings including: warm-start, cold-start item, and cold-start user in terms of Accurcy@k (Yin et al., 2014).

In the following, we first introduce the experimental settings, followed by the experimental results analysis from the above aspects.

### 3.1 EXPERIMENTAL SETTINGS

To evaluate the power of finding out potential users and the accuracy of recommendation in different settings. (1) For the CiteUlike dataset, it contains 5,551 users, 16,980 articles, 204,986 observed user-article binary interaction pairs, and articles abstract content. Similar to (Wang & Blei, 2011), we extract bag-of-the-words item vector with dimension $d_v = 8000$ by ranking the TF-IDF values. (2) For the RecSys 2017 Challenge dataset, it is the only publicly available datasets that contains both user and item content data enabling both cold-start item and cold-start user recommendation. It contains 300M user-item interactions from 1.5M users to 1.3M items and content data collected from the career oriented social network XING (Europern analog of LinkedIn). Like (Volkovs et al., 2017), we evaluate all methods on binary rating data, item content with dimension of $d_u = 831$ and user content with the dimension of $d_v = 2738$. user features and 2738 item features forming the dimensions of user and item content .

We randomly split the binary interaction (rating) $R$ into three disjoint parts: warm start ratings $R^w$, cold-start user ratings $R^u$, and cold-start item ratings $R^v$, and $R^w$ is furtherly split into the training dataset $R^{wt}$ and the testing dataset $R^{we}$. Correspondingly, the user and item content datasets are split into three disjoint parts. The randomly selection is carried out 5 times independently, and we report the experimental results as the average values.

### 3.2 EVALUATION METRIC

The ultimate goal of recommendation is to find out the top-$k$ items that users may be interested in. Accuracy@$k$ was widely adopted by many previous ranking based recommender systems (Koren, 2008; Chen et al., 2009). Thus we adopt the ranking-based evaluation metric Accuracy@$k$ to evaluate the quality of the recommended item ranking list.

**Metric for Marketing Application.** For a new application of the recommender system, there haven't yet a metric to evaluate the marketing performance. Thus, in this paper, we define an evaluation metric similar to the ranking-based metric Accuracy@$k$ used for the warm-start and cold-start recommendation in this paper.

From Fig. 2, we discover the $k$ nearest potential users for an item $j$. The basic idea of the metric is to test whether the user that really interested in an item appears in the $k$ potential users list. For each positive rating ($r_{ij} = 1$) in the testing dataset $D_{test}$: (1) we randomly choose 1000 negative users (users $k$ with $r_{kj} = 0$) and find $k$ potential users in the 1001 user set; (2) we check if the positive user $i$ (with positive rating $r_{ij} = 1$) appears in the $k$ potential users list. If the answer is 'yes' we have a 'hit' and have a 'miss' otherwise.

The metric also denoted by Accuracy@$k$ is formulated as:

$$\text{Accuracy@k} = \frac{\#hit@k}{|D_{test}|}, \tag{12}$$

---

[1] http://www.citeulike.org/faq/data.adp
[2] http://www.recsyschallenge.com/2017/

where $|D_{test}|$ is the size of the test set, and $\#hit@k$ denotes the number of hits in the test set.

### 3.3 ACCURACY FOR MINING POTENTIAL USERS

The experiments evaluate the performance of the marketing application in mining potential users for warm-start items on the test dataset $\boldsymbol{R}^{we}$ and cold-start items on $\boldsymbol{R}^{v}$. Specifically, we first train the model on the training dataset $\boldsymbol{R}^{wt}$ and the corresponding user and item content data. When the training is completed, we fix parameters and obtain hash codes $\boldsymbol{b}_i$ and $\boldsymbol{d}_j$ by making forward passes. Then we generate $k$ potential users for items in the test dataset by the procedure demonstrated in Fig. 2, and evaluate the quality of the potential users list by Accuracy@k defined in Section 3.2. The

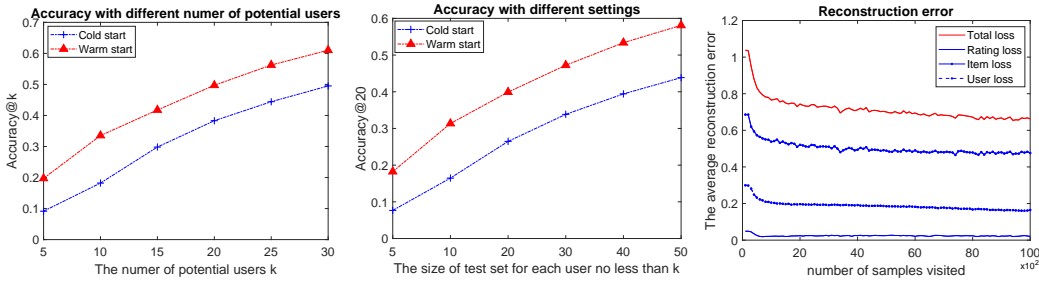

Figure 3: **Left:** The accuracy variation for warm-start and cold-start with the number of potential users. **Center:** The accuracy variation of mining 20 potential users with the number of users (size of the test set) who really interested in the target item. **Right:** The average reconstruction error with the number of visited samples. The total error is the sum of the rating reconstruction error, the user content reconstruction error, and the item reconstruction error.

marketing analysis for warm start item and cold-start item are reported in Fig. 3 (**Left.**), which shows the accuracy values varies with the numbers of potential users. It indicates the accuracy increases with the number of potential users for both cold-start and warm start settings. It's reasonable because mining more potential users will have greater accuracy value defined in Section 3.2. Especially, the proposed CGH is effective for cold-start item, which indicates further e-commerce strategies can be developed for new items to attract those potential users. Besides, from the perspective of marketing, warm-start recommendation and cold-start recommendation has less gap than traditional recommendation.

**Robust Testing.** We evaluate the performance varies with the number of users who really interested in the target item in test set. The experimental results shown in Fig. 3 (**Center.**) indicates the accuracy grows steadily with the size of test set, which reveals the CGH for marketing application is robust. Thus, it is practical to be used in the sparse and cold-start settings.

**Convergence of CGH.** Fig. 3 (**Right.**) demonstrates the convergence of the proposed CGH, which reveals the reconstruction errors of ratings, users content, items content and the total error with the number of samples seen by CGH are converged, which furtherly validate the correction and effectiveness of the proposed CGH.

### 3.4 ACCURACY FOR RECOMMENDATION

**Accuracy for warm-start Recommendation.** Fig. 4 (**Left.**) shows the accuracy comparison of warm-start recommendation on CiteUlike dataset. In which collaborative generated embedding (CGE) denotes the real version of CGH. The figure shows the proposed CGH (CGE) has a comparable performance with other hybrid recommender systems. The proposed CGH is hashing-based recommendation, hence binary vectors apply to recommendation which has the advantage in online recommendation as introduced in Section 1; while the baselines are real-valued recommendations which conducts recommendation on real latent space. Due to real latent vectors intuitively carried more information than hash codes. Thus it is acceptable to have small gaps between the real-valued hybrid recommendation and the hashing-based recommendation. In addition, there is still small gap of the real version CGE in comparison with DropoutNet, because the reconstruction error is consid-

ered in CGH(CGE), while DropoutNet didn't consider it. However, the reconstruction is significant in the generative step of CGH, which makes it feasible to mining effective potential users, thus CGH(CGE) has the advantage in marketing application.

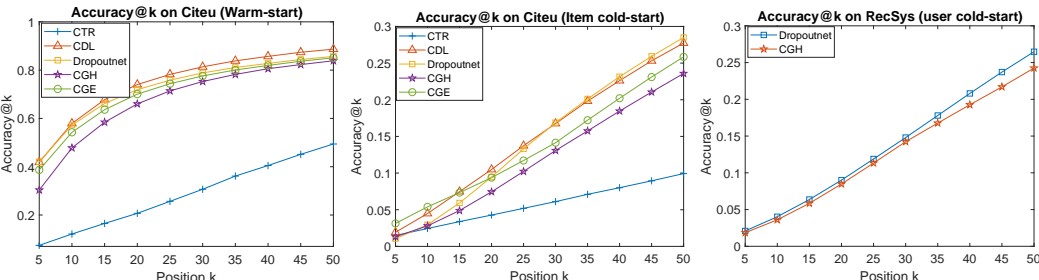

Figure 4: **Left:** The Accuracy variation with the number of recommended items for warm-start recommendation on CiteUlike. **Center:** The Accuracy variation with the number of recommended items for cold-start item recommendation on CiteUlike **Right:** The Accuracy variation with the number of recommended items for cold-start user recommendation on RecSys

**Accuracy for cold-start item recommendation.** This experiment studies the accuracy comparison between competing hybrid recommender systems and CGH under the same cold-start item setting. We test the performance on the test dataset $R_v$ introduced in Section 3.1. Specifically, in $R_v$ each item (cold-start item) has less than 5 positive ratings. Then we select users with at least one positive rating as test users. For each test user, we first choose his/her ratings related to cold-start items as the test set, and the remaining ratings as the training set. Our goal is to test whether the marked-off cold-start items can be accurately recommended to the right user.

The experimental results for cold-start item recommendation are shown in Fig. 4 (**Center.**). We conclude that CGH has a comparable performance with competing baselines and achieves better performance than CTR. The results evaluated by another metric MRR (detailed in Appendix.A) are similar.

**Accuracy for cold-start user recommendation.** We also test the performance on the cold-start user setting on the test dataset $R_u$ introduced in Section 3.1. Specifically, in $R_u$, each user (cold-start user) has less than 5 positive ratings. Then we select items with at least one positive rating as test items. For each test item, we first choose ratings related to cold-start users as the test set, and the remaining ratings as the training set. Our goal is to test whether the test item can be can be accurately recommended to marked-off user.

Due to the fact that only Dropout can be applied to cold-start user recommendation, so we only compare the performance of CGH with Dropout. The experimental results for cold-start user recommendation shown in Fig. 4 (**Right.**) indicates our proposed CGH has similar performance with DropoutNet. Besides, CGH has the advantage of the application in marketing area.

## 4  CONCLUSION

In this paper, a generated recommendation framework called collaborative generated hashing (CGH) is proposed to address the cold-start and efficiency issues for recommendation. The two main contributions are put forward in this paper: (1) we develop a collaborative generated hashing framework with the principle of Minimum Description Length together(MDL) with uncorrelated and balanced constraints on the inference process to derive compact and informative hash codes, which is significant for the accuracy of recommendation and marketing; (2) we propose a marketing strategy by the proposed CGH, specifically, we design a framework to discover the $k$ potential users by the generate step; (3) we evaluate the proposed scheme on two the public datasets, the experimental results show the effectiveness of the proposed CGH for both warm-start and cold-start recommendation.

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

## APPENDIX

### A. MRR RESULTS ON FOR RECOMMENDATION

We evaluate the accuracy in terms of the MRR (Yin et al., 2014) metric shown in Table 1 for warm-start recommendation. Our proposed CGH performs almost as well as the best result of the real-valued competing baselines. Table 1 summarizes MRR results for the four algorithms, the best result is marked as '$\star$q  and the second best is marked as '$o$q . We find that the performance of CGH is very close to the best result, that is consistent with the outcome of Accuracy@$k$ reported in Fig. 4.

Table 1: MRR on CiteUlike

| Method | CTR | CDL | Dropoutnet | CGH |
|---|---|---|---|---|
| 'Warm-startq | 0.0324 | 0.0667$^\star$ | 0.0580 | 0.0595$^o$ |
| 'Cold-startq | 0.0101 | 0.0150 | 0.0179$^\star$ | 0.0165$^o$ |

