# OpenReview forum: "Collaborative Generated Hashing for Market Analysis and Fast Cold-start Recommendation"
_ICLR.cc/2020/Conference — Reject_

### Official Review · AnonReviewer1 · 2019-10-06
**Official Blind Review #1**

**Rating:** 1

**Review:**

The paper proposes new hashing schemes to learn hash codes to describe users/items for the purpose of recommendation.

It is claimed that doing so leads to three contributions
(1) the new hash codes themselves, which can apply to various recommendation settings
(2) the ability to better discover potential users in e-commerce settings
(3) state-of-the-art performance on several datasets

In terms of (1), the paper contrasts with Collaborative Deep Learning and dropoutnet. The main difference compared to CDL is that the hashing-based method learns binary codes. Compared to DropoutNet the main difference is the use of a stacked autoencoder, rather than a different neural network architecture. This latter contribution is perhaps a little thin (really it's just a technical detail); the additional contribution of being able to be used as a marketing strategy I didn't really follow.

The "mining potential users" contribution (contribution 2) seemed a little ad-hoc to me. It ultimately seems like a variant of KNN, and seems like something similar could be attempted for other methods.

Training etc. looks fine, though I didn't fully check the details.

The experiments seem not totally convincing. Most critically, the method does not seem to exhibit state-of-the-art performance as claimed, but is somewhat lower (in terms of accuracy) than other baselines. It might be better in terms of speed but this doesn't seem to be thoroughly evaluated. The choice of accuracy as the only evaluation metric also seems unusual.

The selection of datasets is also quite limited. It's claimed that these are the only datasets with user and item content, but why are both needed to run an experiment? Can't this method work with either (in which case many other datasets would be appropriate)?

Overall the actual results seem mixed, and thus the paper hinges on its statement that it has the "advantage of applications in marketing area". However this latter contribution seems handwavy.

In order to be accepted, I'd need to see
-- More clearly stated and demonstrable contributions
-- More compelling experiments, in terms of datasets, evaluation measures, and actual performance

**Experience Assessment:**

I have published in this field for several years.

**Review Assessment: Checking Correctness Of Derivations And Theory:**

I did not assess the derivations or theory.

**Review Assessment: Checking Correctness Of Experiments:**

I assessed the sensibility of the experiments.

**Review Assessment: Thoroughness In Paper Reading:**

I made a quick assessment of this paper.

---

### Official Review · AnonReviewer3 · 2019-10-22
**Official Blind Review #3**

**Rating:** 1

**Review:**

This paper introduces a collaborative generated hashing (CGH) method to learn hash funcations of users and items from content data. The approach first provides a strategy to discover potential users by the generative step and inference through adding balanced and uncorrelated constraints. The experiments demonstrates some effectiveness on improving accuracy for both warm-start and cold-start recommendations.

This paper should be rejected because (1) this method only combines existing techs, such as Stochastic Generative Hashing (Eq.1 and Eq. 6), and lacks novelty; (2) lack of introduction to related work and baselines, (3) the experiments results can not support the claim, i.e. the effectiveness of CGH in marketing area, and (4) paper writing is awful and very hard to follow.

Main argument

Almost every essential parts of the proposed method are from existing methods:
(I) Eq. 1 and Eq. 6 are proposed by Stochastic Generative Hashing [1];
(II) Eq. 2 and Eq. 5 are a multivariate Bernoulli distribution;
(III) Eq. 3 is a normal distribution;
(IV) Eq. 7 is proposed by [2];
(V) loss function Eq. 9 is follow the Minimum Description Length principle [1];
The proposed method CGH is a combination of these techs and compared with these methods, there are few novel aspects.

This paper omits the related work part and does a rough introduction to two baselines (CDL and DropoutNet) in a confusing way in Section 2. A concise and precise introduction to other methods will help the reader to better understand the related works and the advantages and disadvantages of the proposed method.

The experiments do not provide convincing evidence of the corretness of the proposed method, especially in Section 3.3. In Section 3.3, Figure 3 shows the performance on Accyracy@k without any baseline. The results do not demonstrate the validity of the method and therefore cannot support the author's claim.

Things to improve the paper that did not impact the score:
1) page 1, 4th line in the 3rd paragraph, 'efficient' -> efficiently
2) page 3, 1st sentence in Section 2.1
3) page 3, hard to find the definition of the similarity function
4) page 4, 2nd line 'similar for' -> 'similar to'
5) page 6, 3rd paragraph 'From Fig. 2, we discosver the k nearest potential users for an item j'. What do you mean?


Reference
[1] Bo Dai, Ruiqi Guo, Sanjiv Kumar, Niao He, and Le Song. Stochastic generative hashing. In Proceedings of the 34th International Conference on Machine Learning-Volume 70, pp. 913–922. JMLR. org, 2017.
[2] Ke Zhou and Hongyuan Zha. Learning binary codes for collaborative filtering. In Proceedings of KDD’12, pp. 498–506. ACM, 2012.

**Experience Assessment:**

I have published in this field for several years.

**Review Assessment: Checking Correctness Of Derivations And Theory:**

I carefully checked the derivations and theory.

**Review Assessment: Checking Correctness Of Experiments:**

I carefully checked the experiments.

**Review Assessment: Thoroughness In Paper Reading:**

I read the paper thoroughly.

---

### Official Review · AnonReviewer2 · 2019-10-22
**Official Blind Review #2**

**Rating:** 1

**Review:**

The work considers the problem of efficient user and item recommendations in the warm- and cold-start settings. It aims at improving computational efficiency of the best candidate selection in these settings by utilizing binary codes representation. The transformation from an actual to a binary code representation is learned in a hybrid manner using both collaborative and content information. In order to keep such representations compact yet expressive enough, the authors impose a set of constraints that ensure balanced and uncorrelated transformations. Once binary codes are learned, the inference can be made by virtue of efficient Hamming distance computations. Moreover, the search for candidate entities can be performed via the generative step that projects binary codes onto actual feature space, where kNN-based techniques can be further utilized.

Major drawback of the work is that it does not provide any quantitative evidence to support the main claim – that the proposed approach is at least more computationally efficient, since it underperforms competing methods in terms of accuracy. Essentially, the work answers the question whether it is possible to utilize hashing techniques based on binary codes; however, the question on the practicality and efficiency of this approach remains open. I would therefore suggest rejecting the work.

One of the weak points of other methods noted by the authors is “the expensive similarity search in real latent space”. The authors aim to resolve that problem by learning hashing functions based on “compact and informative” binary codes representation. However, while an overall problem formulation is clearly described and the learning objective is well explained, no further evidence supporting initial claims is provided. Moreover, an overall logic seems contradictory:
1.	Binary codes allow efficient preference estimation via XOR operation.
2.	Learning binary codes is a difficult discrete optimization task.
3.	Hence, we employ special MDL principle for solving a constraint optimization problem and employ relaxation of hash codes to move away from discrete optimization.
After relaxation, the hash codes are no longer binary. Do you still enforce binary representation by some thresholding or other method? If yes, more words explaining this should be added to the text. If no, then how is it different from classical learning of latent variables? Essentially, relaxed non-binary hash codes are similar to latent vectors.
The lack of description raises concerns in an overall efficiency and the authors, unfortunately, provide no evidence of improved computational performance. Given that the proposed approach underperforms competing methods in terms of accuracy of recommendations, more efforts should be made to demonstrate its competitive advantages in terms of time required for training and generating predictions.

Another contradictory part is the generative step for candidate selection. The idea of using inference to generate the most pertinent user vector for a selected item hash code is novel and interesting. However, it requires searching neighbors in the real feature space, which can be very inefficient, depending on the structure of features. I’m not convinced that it is better than searching neighbors directly in the latent space, which can be done in the majority of hybrid models. Moreover, there exist various approximate nearest-neighbors search methods, e.g. Annoy, NMSLib, Faiss, etc., which allow trading-off accuracy and efficiency. Considering that hash codes also lose some information (which is observed in the results of experiments), it seems necessary to have a comparison with these approximate methods as well.
It should be also noted that in some cases you don’t even have to run the similarity search. Many hybrid models learn latent representation of features directly and cold-start entities are straightforwardly described via combination of the corresponding latent vectors of their features (e.g, Factorization Machines [Rendle 2009]). Hence, affinity between a cold item and some user can be quickly estimated via inner product of their latent vectors.

Suggestions on improving the work:
Worth mentioning, that recent studies raise certain concerns about the superiority of modern neural network-based approaches over simpler (and properly tuned) linear baselines, see the work by [Dacrema, Cremonesi, Jannach 2019] on “A Worrying Analysis of Recent Neural Recommendation Approaches”. The DropoutNet method in your experiments is very similar to CDL in terms of accuracy. The latter, however, underperforms even simple knn models, as shown by the work mentioned above. The haven’t tested it in the cold-start regime, though. Still, I’d strongly recommend adding to your experiments comparison with simpler hybrid models, e.g., Factorization Machines. Also note that there are even stronger baselines published recently, e.g. HybridSVD by [Frolov, Oseledets 2019].

Additional remarks:
1)	Figure 3 and the related text seem to focus on too obvious things. Indeed, by increasing the number of entities to compare against, you increase chances to have a hit. This part of the text, basically, states that the method works, which can already be seen from other results.
2)	A lot of attention is given to the “marketing application”. It’s ok to have it in introduction and make a connection to the real-world problem; however, further mentions of it in the text feel excessive. In the experiment section you describe a standard evaluation procedure for the cold start, there is no need to refer to marketing application again as you do not provide any new metric. It would feel much more organic if you would have the results of A/B testing on real users. Otherwise, I’d suggest to focus more on the problem that you’re solving, not on possible application.
3)	The text is in a very unpolished state. It reads more like a draft version. There are many typos and error both in text and in derivations.

References:
Rendle, Steffen. "Factorization machines." In 2010 IEEE International Conference on Data Mining, pp. 995-1000. IEEE, 2010.
Frolov, Evgeny, and Ivan Oseledets. "HybridSVD: when collaborative information is not enough." In Proceedings of the 13th ACM Conference on Recommender Systems, pp. 331-339. ACM, 2019.
Dacrema, Maurizio Ferrari, Paolo Cremonesi, and Dietmar Jannach. "Are we really making much progress? A worrying analysis of recent neural recommendation approaches." In Proceedings of the 13th ACM Conference on Recommender Systems, pp. 101-109. ACM, 2019.

**Experience Assessment:**

I have published in this field for several years.

**Review Assessment: Checking Correctness Of Derivations And Theory:**

I assessed the sensibility of the derivations and theory.

**Review Assessment: Checking Correctness Of Experiments:**

I assessed the sensibility of the experiments.

**Review Assessment: Thoroughness In Paper Reading:**

I read the paper thoroughly.

---

### Decision · Program_Chairs · 2019-12-19

**Decision:**

Reject

**Comment:**

The presented work has worse accuracy than existing (and not all the baselines are given correctly) and does not provide the running time comparison. All reviewers recommend rejection, and I am with them.